# OpenReview forum: "Mitigating Object Hallucination in Large Vision-Language Models via Image-Grounded Guidance"
_ICML.cc/2025/Conference — ICML 2025 spotlightposter_

### Official Review · Reviewer_GyxS · 2025-03-06

**Overall Recommendation:** 3

**Summary:**

This paper proposes the MARINE framework to address the object hallucination issue in Large Vision-Language Models (LVLMs). This framework introduces visual guidance from image-grounded models to effectively reduce hallucinations during inference. Experiments show that MARINE outperforms baseline methods on multiple LVLMs and balances latency and accuracy.

**Claims And Evidence:**

Most claims are supported by clear evidence. The authors conduct extensive experiments on five LVLMs using multiple metrics like CHAIR, POPE, and GPT-4V-aided evaluation. They compare MARINE with various baselines and perform ablation studies.
However, the experiments are limited to specific datasets and tasks, which can be problematic. I will give my reason in the Methods And Evaluation Criteria section.

**Essential References Not Discussed:**

I think most of the listed related works are essential for understanding this paper's contributions. While there are some other works in Large vision language models (LVLMs) that seem to apply feature steering to mitigate hallucinations, such as:

[1] Reducing hallucinations in vision-language models via latent space steering, 2024.

[2] Nullu: Mitigating Object Hallucinations in Large Vision-Language Models via HalluSpace Projection, 2024.

These are suggested to be discussed in related works.

**Experimental Designs Or Analyses:**

See Evaluation Criteria.

**Methods And Evaluation Criteria:**

Overall, the paper is well-written and easy to follow. The technical routing makes sense to me. My primary concern is implementing the vision models, such as DETR. As far as I know, both CHAIR and POPE contain the samples selected from MS COCO, and using the DETR  trained on MS COCO can surely improve the method's performance. In this case, the results on CHAIR and POPE can be somehow unfair. Note that the compared method, VCD, does not introduce additional information and achieve comparable performance.

Can the author provide more explanation about this point?

**Other Comments Or Suggestions:**

I am willing to adjust the score if the issues are satisfactorily addressed.

**Other Strengths And Weaknesses:**

Generally, the paper is well-written and easy to follow, and the results seem to be good.

Using 2D features from pre-trained models to prompt downstream tasks such as 3D detection and 2D few-shot detection is not a new concept. It would be beneficial to have a more in-depth explanation of how this method differs from direct prompting.

**Questions For Authors:**

None.

**Relation To Broader Scientific Literature:**

The paper's key contribution to MARINE in mitigating object hallucination in LVLMs builds on using knowledge from vision models. MARINE offers a training-free and API-free approach. By leveraging image-grounded models, the root causes of hallucination are addressed well.

**Theoretical Claims:**

The theoretical claims are right.

---

> ### Author Rebuttal · Authors · 2025-04-01
>
> Thank you for your insightful feedback and acknowledgement of our extensive experiments and the overall clarity and structure of our paper. We detail our response as follows.
>
> ### Q1: Using the DETR trained on MSCOCO may be unfair.
> MSCOCO (train) is a widely-used open-source image-caption dataset frequently utilized for pre-training various vision encoders, including CLIP, the backbone of current LVLMs (e.g., LLaVA). While all vision encoders leverage data that includes MSCOCO training split, the difference in how to process and leverage these information determines model performance. This variability is evident in MARINE’s superior performance compared to LVLMs utilizing only the CLIP encoder. Additionally, our manuscript's Table 6 illustrates instances where MARINE with only DETR underperforms compared to MARINE employing only RAM++. Nonetheless, aggregating information from multiple visual encoders (as in MARINE) consistently achieves the highest performance.
>
> Moreover, the vocabulary derived from MSCOCO effectively encompasses frequent objects common across diverse natural-image datasets, demonstrating strong generalizability to other evaluation data. **This is confirmed by the substantial improvements in POPE results on A-OKVQA shown in Table 15**, highlighting our method's capability to generalize effectively and outperform baselines beyond MSCOCO.
>
> ### Q1.1: Comparison with VCD.
> We acknowledge that VCD is a valuable work, but our approach addresses the hallucination problem from a different and effective perspective. VCD aims to reduce LVLMs’ over-reliance on language priors from LLM pre-training data by contrasting distorted visual inputs. In contrast, MARINE focuses on hallucinations arising from insufficient visual context. Thus, VCD and MARINE approach hallucination from complementary angles, and integrating both methods has the potential to achieve further performance improvements.
>
> ### Q2: Relevant works on feature steering for hallucination reduction.
> Thank you for pointing to these relevant lines of research on feature steering. We summarize them as follows and will include the discussion in our next revision.
>
> Nullu identifies a “HalluSpace” by comparing truthful and hallucinated features, then projects model weights to the null space of those hallucination-prone directions, reducing object hallucinations with no extra runtime cost. VTI (Visual and Textual Intervention) learns “shift vectors” by analyzing how vision features change under corruption and how text features differ between hallucinated and correct outputs. It then applies these shifts at inference to stabilize LVLMs and reduce hallucinations.
>
> ### Q3: More in-depth explanation of how this method differs from direct prompting.
>
> We clarify key conceptual and empirical differences between MARINE and prompting-based methods, which we believe directly address this concern:
> - Direct prompting relies solely on the model’s textual instruction-following capabilities, which can exacerbate hallucination issues in less capable models (e.g., LLaVA). Conversely, MARINE directly enhances visual understanding by integrating additional visual information, making it effective even for weaker models without incurring training overhead.
> - Prompting methods cannot introduce new visual data and thus remain constrained by the original vision model’s capabilities. MARINE introduces novel visual information through an improved vision encoder, fundamentally enhancing the model's observational accuracy.
> - Direct prompting and additional prompt methods require careful crafting and tuning specific to each task or dataset. MARINE, however, exhibits strong generalization capabilities across diverse models and datasets, effectively reducing hallucinations and improving reliability without additional training or manual prompt optimization.
>
> Appendix B.1.1 further elaborates on these distinctions. In particular, we evaluated the direct prompting baseline using a highly detailed instruction explicitly guiding the model to describe only observable visual characteristics
>
> As shown below, MARINE consistently outperforms the prompting baseline by significantly reducing hallucinations while maintaining or improving recall. Although prompting can improve recall in some cases, it often worsens hallucination metrics. MARINE achieves better overall reliability across all models.
>
> | Method  | LLAVA-$C_s\downarrow$ | LLAVA-$C_i\downarrow$ | LLAVA-$Recall\uparrow$ | LLaVA-v1.5-$C_s\downarrow$ | LLaVA-v1.5-$C_i\downarrow$ | LLaVA-v1.5-$Recall\uparrow$ | mPLUG-Owl2-$C_s\downarrow$ | mPLUG-Owl2-$C_i\downarrow$ | mPLUG-Owl2-$Recall\uparrow$ |
> |-----|------|------|-----|------|-----|-----|-----|-----|------|
> | Original  | 26.6  | 10.5 | 47.4 | 8.8 | 4.6 | 41.1 | 6.2 | 3.4 | 38.8 |
> | Direct Prompting | 27.2| 11.0| 46.4| 19.6 | 8.3  | **52.3**  | 9.0  | 5.1  | **42.0** |
> | **MARINE** (ours)  | **17.8**  | **7.2**  | **50.8**  | **6.2** | **3.0** | 44.3| **4.2** | **2.3** | 41.4|

---

### Official Review · Reviewer_R69G · 2025-03-09

**Overall Recommendation:** 3

**Summary:**

The paper proposes the MARINE method for mitigating object hallucination in LVLMs. The method uses results from external object detection models and adds it in the form of an extra textual prompt into the LVLM’s generation. The method is compared with several baselines on object hallucination benchmarks, as well as on VQA and image captioning tasks. Several ablation studies are also presented in the paper.

**Claims And Evidence:**

The paper makes the following claims:

1. “MARINE mitigates insufficient visual context provided by the visual encoder and misalignment between the vision and text domains”: I think this claim is too general. First, MARINE cannot be used with an arbitrary visual encoder but rather with object detection models. Second, it is unclear to me where in the experimental section the misalignment between the vision and text domains is investigated and which experiments demonstrate that MARINE mitigates it.

2. “MARINE does not require additional training resources or access to advanced LLMs.”: This is true but it instead requires access to advanced object detection models, which should be made more clear in the paper.

3. “MARINE outperforms the baselines in hallucination mitigation while maintaining overall performance across multiple tasks (image captioning, VQA)”: I believe that this claim has been demonstrated in the experimental section, though there are cases where MARINE does not outperform the baselines (eg in Table 1 and 2).

**Essential References Not Discussed:**

See my comment above about SILC: Improving Vision Language Pretraining with Self-Distillation, ECCV 2024 and BRAVE : Broadening the visual encoding of vision-language models, ECCV 2024.

**Experimental Designs Or Analyses:**

1. In Appendix, you state that “For decoding methods such as VCD, OPERA and our method, we measured the latency of LLaVA generating captions directly”. I think this is unfair, as MARINE additionally requires forward passes through multiple external models which in my opinion should be counted towards the latency calculation.

2. See also my comment above regarding the misalignment analysis (under claims).

**Methods And Evaluation Criteria:**

While the chosen benchmarks make sense, it would be great to compare MARINE against more powerful VLMs that similarly incorporate more fine-grained visual information, like SILC: Improving Vision Language Pretraining with Self-Distillation, ECCV 2024 and BRAVE : Broadening the visual encoding of vision-language models, ECCV 2024. Especially BRAVE is conceptually similar to MARINE in that it incorporates information from multiple encoders.

**Other Comments Or Suggestions:**

Please label y axis and change colors in Figure 3, 8, 14 and 15 as the lines are hard to distinguish.

**Other Strengths And Weaknesses:**

1. The novelty of the paper is a bit limited and the claims made in the introduction are too general. In particular, I would like the authors to rephrase the paper and remove any occurrences where it is claimed that information from a general visual encoder is used because this is not true (at least it is not shown in the paper). In addition, MARINE is only useful when used on images where the pretrained object detectors give meaningful outputs, which should also be stated in the paper.

2. Despite the limited novelty, I find the simplicity of the method and extensive experimental evaluation a strength.

**Questions For Authors:**

Please see my questions in the sections above.

**Relation To Broader Scientific Literature:**

Mitigating object hallucinations is an active area of research.

**Theoretical Claims:**

I do not understand what MARINE-Truth is and neither can I find any details in Appendix A as stated in the main paper. I also do not understand the point of this part. Please explain in the main paper what it is and provide a better explanation on why it is important to look at it.

---

> ### Author Rebuttal · Authors · 2025-04-01
>
> Thank you very much for your insightful review and valuable feedback. We sincerely appreciate your recognition of the simplicity and thorough experimental validation of our MARINE approach.
>
> ### Q1.1 Clarify the claim regarding misalignment.
> In the original claim, by "visual encoder," we referred to the LVLM's visual encoder component (e.g., CLIP), highlighting that MARINE supplements its initially insufficient visual context. The mentioned misalignment issue refers that LVLMs typically employ a trainable linear alignment layer, which can potentially lose or distort important object-level information during the mapping from visual to textual representations. In contrast, MARINE directly extracts and utilizes explicit object-level details, thereby preserving critical visual information intact. We will clarify this point explicitly in the revised manuscript.
>
> ### Q1.2. Clarify the claim regarding access to advanced models.
> Thank you for pointing this out. We will clarify that MARINE inherently requires access to additional vision models to enrich the visual inputs in our revision. Additionally, the "advanced LLMs" that we referenced refer to closed-source models that are accessible only via paid APIs (e.g., GPT-4o), while the vision models are open-source. We will explicitly state this distinction in our revision.
>
> ### Q2. Could you explain the concept and importance of "MARINE-Truth" clearly in the main paper?
> We consider using the ground-truth object list as a variant of MARINE and denote it as MARINE-Truth. The performance of MARINE-Truth serves as a reference to MARINE’s best performance. However, the ground-truth object list may also contain noise and therefore occasionally underperforms MARINE. We will explain this in our revision.
>
> ### Q3. In your latency measurements, why did you exclude the latency of additional forward passes through external models required by MARINE?
> Thank you for raising this point. To clarify, we did include the latency of additional forward passes through external vision models in our measurements. These external vision models have negligible inference overhead compared to autoregressive models (e.g., LLMs, LVLMs). To illustrate clearly the impact of including or excluding the latency introduced by these external models, we present a detailed latency comparison in the table below. Specifically, the table includes scenarios both excluding visual prompt generation latency (Offline MARINE) and including it (Online MARINE):
> |   | Greedy  | LURE  | Woodpecker*  | VCD  | OPERA | **Offline MARINE**   | **Online MARINE**  |
> |------|-----|-----|-----|-----|------|------|------|
> | **Training Cost** | 0  | 10min on A100 80G | 0  | 0 | 0 | 0 | 0  |
> | **Inference Latency (ms/token)** | 26.3 (×1.0) | 179.9 (×6.84) | 94.5 (×3.59)* | 53.4 (×2.03) | 185.1 (×7.0) | **52.2 (×1.98)** | **52.23 (×1.985)** |
> *Woodpecker requires GPT API key access, and the latency may depend on OPENAI API.
>
> As shown, processing images via additional vision models adds only a negligible overhead to the overall latency.
>
> ### Q4. Can you compare MARINE against other SOTA VLM methods, especially BRAVE?
> Thank you for pointing out these related works, we will include them in discussion in our next revision. Especially, BRAVE indeed shares a very similar intuition as ours for ensembling diverse visual information sources to improve model faithfulness, confirming the motivation of our work. Here, we compare MARINE against BRAVE on the POPE benchmark. As shown below, MARINE achieves comparable performance than BRAVE, while introducing no additional trainable parameters.
>
> | Model    | Total Params | Trainable Params | Rand  | Pop   | Adv   | POPE$_\text{avg}$ |
> |--------------|--------------|------------------|-------|-------|-------|-------------------|
> | LLaVA-v1.5 (7B)   | 7B  | 7B      | 87.3  | 86.1  | 84.2  | 85.9|
> | BRAVE    | 10.5B| 3B      | –     | –     | –     | 87.6     |
> | **MARINE (ours)** | 7B       | 0  | **87.9** | **86.5** | **86.7** | 87.0 |
>
> *Note: We report BRAVE’s POPE$_\text{avg}$ score as stated in their paper. Their model and detailed evaluation results have not been open-sourced. For fair comparison, we adopt the same test set as LLaVA-v1.5.*
>
> ### Q5: Labels in figures.
> Thank you for catching these. We will update them in our next revision.

---

### Official Review · Reviewer_SDtF · 2025-03-14

**Overall Recommendation:** 4

**Summary:**

The paper presents a novel method called MARINE to reduce hallucination in large vision-language models (LVLMs).

The method can be applied to LVLMs without any training. When auto-regressively generating individual tokens, logits are computed twice: once with the normal LVLM input ("unconditional"), and once with an augmented "conditional" input containing tokens from visual guidance models (DETR and RAM++). The logits of the unconditional and conditional inputs are then combined for sampling the next token ( similar to classifier-free guidance in diffusion models).

The paper compares the method with 5 LVLMs against 5 different baselines and shows improvements of both CHAIR and POPE scores, as well as an improvement of caption metrics. MARINE is also less compute intensive than the baselines methods.

**Claims And Evidence:**

1. The paper claims that the MARINE method is effective for mitigating object hallucinations in LVLMs.

The paper gives good evidence that the method indeed works well compared to previous methods.

Theoretically, the method is based on classifier-free guidance, which was introduced in diffusion models (Ho and Salimans, 2021) and then adapted for text sampling (Sanchez, 2023).

1a. The paper always uses greedy sampling, and previous works such as (Sanchez, 2023) have sampled at different temperatures. The paper simply assumes it works equally well with greedy sampling, without referring to previous literature, experimental evidence, or theoretical considerations.

1b. In Section 5.3 (Line 428) the paper states that the best guidance strength is between 0.3 and 0.7. Figure 3 shows that object hallucinations decrease with increasing guidance strength for LLaVA, but not for for mPLUG-OWL2. It would be interesting to see how this affects other models, maybe mPLUG-Owl2 is an outlier. Figure 8 in the appendix shows that increased guidance strength improves captioning metrics on captioning tasks (I'm less convinced of the value of the captioning metrics on the LLaVA-QA90 task). Then the only evidence for guidance strength 0.7 being better than guidance strength 1.0 is Table 22 in the appendix. This seems somewhat scarce evidence for this central parameter (note that with guidance strength 1.0 the method reduces to something much simpler).

1c. Ιn section 5.3 the paper claims that "intersection-based method outperforms the union" and this is based on Table 6. I find it surprising that using a single model only *increases* sentence-level hallucinations for LLaVA models. Note that Table 9 in the appendix lists different prompts for MARINE-intersec and MARINE-union. Which models to use and how to combine their output is also a central part of the method, so I think this should be explained in some more detail with better evidence.

2. The paper claims that MARINE is training-free and compares favorably with existing methods. This claim is well supported by the fact that MARINE can be applied to a variety of models without re-training (Tables 1,2) and by the inference latency measurements (Table 5).

**Essential References Not Discussed:**

None.

**Experimental Designs Or Analyses:**

The method is compared with four previous methods (LURE, Woodpecker, VCD, OPERA – all from 2023). This evaluation seems robust and fair.

**Methods And Evaluation Criteria:**

The benchmark datasets are appropriate. The method is evaluated on CHAIR and POPE (original MSCOCO, as well as A-OKVQA and GQA) to measure the mitigation of hallucinations.

Additionally, the method is evaluated on and GPT-4V-aided evaluation (Yin, 2023), which also measures hallucinations, but at the same time gives information about the usefulness of the outputs.

Finally, the method is also evaluated on captioning metrics, again with the goal to verify that the output quality other than hallucinations does not deteriorate.

**Other Comments Or Suggestions:**

Typos:
1. Line 102 (right column): "studies(Li"
1. Line 137: "2023a)and"
1. Line 223: "InstructBLIP (Liu et al., 2023c)"
1. Line 382 (right column): experimental evidence for guidance strength is provided in Appendix B.4 (not C)
1. Line 682: "ROUGH-L"
1. Lines 360-362: missing spaces before "("

Questions:
1. Line 262 (right column): what is the "noise intensity" for DETR?
1. Line 264 (right column): why only greedy sampling? in (Sanchez, 2023) different sampling temperatures were explored, but the method was never applied with greedy sampling
1. Lines 716-748 (Table 9): why use different prompts in MARINE-intersec and MARINE-union?

Nits:
1. Lines 289: consider mentioning that these results are on MSCOCO – this would make it clearer how Table 2 relates to Table 4
1. Line 1110: awkward placement of orphan line

**Other Strengths And Weaknesses:**

Strengths and weaknesses are discussed in above sections.

**Questions For Authors:**

See above sections.

**Relation To Broader Scientific Literature:**

The work is well anchored in existing literature about object hallucinations in LVLMs and controllable generation.

Section 2.1 mentions some more recent work from 2024, but all baselines (Section 5.1) are from 2023. Why is the work not compared to newer methods?

**Theoretical Claims:**

There are not theoretical claims in the paper. Note above comments about greedy sampling and guidance strength.

---

> ### Author Rebuttal · Authors · 2025-04-01
>
> Thank you very much for your thoughtful and constructive review. We appreciate your recognition of MARINE’s effectiveness and comprehensive evaluation. We provide detailed responses to your questions below:
>
> ### Q1. Effect of sampling temperatures.
> In our paper, we opted for greedy sampling (temperature = 0) to ensure deterministic behavior in LLMs, eliminating randomness and thereby facilitating more reliable comparisons. This setup is also consistent with our main baseline VCD, and is widely used in existing benchmark evaluations to ensure reproducibility.
>
> Here, we conducted experiments using temperature = 0.6 and reported the mean ± standard deviation in the table below. As shown, MARINE consistently improves object hallucination metrics regardless of the sampling strategy.
>
> | Method | LLAVA-$C_s\downarrow$  | LLAVA-$C_i\downarrow$  | LLAVA-$Recall\uparrow$  | mPLUG-Owl2-$C_s\downarrow$  | mPLUG-Owl2-$C_i\downarrow$ | mPLUG-Owl2-$Recall\uparrow$ |
> |----|----|----|-----|-----|-----|-----|
> | Original | 26.1 ± 1.6  | 10.8 ± 0.5  | 46.0 ± 0.8 | 4.9 ± 0.6   | 2.8 ± 0.3 | 37.7 ± 0.6   |
> | MARINE (ours) | 19.3 ± 0.8$_{-6.8}$  | 7.6 ± 0.1$_{-3.2}$ | 50.6 ± 0.2$_{+4.6}$ | 4.5 ± 0.6$_{-0.4}$     | 2.4 ± 0.2$_{-0.4}$ | 41.1 ± 0.4$_{+3.4}$ |
>
> ### Q2. Questions on guidance strength. Is mPLUG-OWL2 an outlier in Figure 3?
> Increasing guidance strength generally improves model faithfulness across all evaluated models, indicated by a notable decrease in the CHAIR. However, the optimal guidance strength varies by model. mPLUG-OWL2 serves as example of the more advanced models that begin with an inherently lower CHAIR score, suggesting it captures visual information better than LLaVA. Thus, it benefits from guidance strengths other than strictly 1.0, where the visual guidance effectively complements rather than dominates its internal visual encoder. In contrast, earlier models like LLaVA rely significantly more heavily on the introduced visual guidance, as it can largely dominate their intrinsic visual grounding capabilities.
>
> However, excessively strong visual guidance can overall harm a model’s ability to follow instructions accurately. This negative effect is illustrated in Figures 9 and 13 of our manuscript, where a guidance strength of 1 reduces the quality of model generations for tasks beyond visual grounding.
>
> For simplicity and consistency, we adopted a universal strength of 0.7 across experiments. Nonetheless, tuning this hyperparameter for each base LVLM using a validation set could yield optimized results tailored to each model. We will include this discussion in our next revision.
>
> ### Q3.1. Why the intersection-based method outperforms the union-based method?
> The intersection-based method retains only visual signals consistently grounded across different vision models, while the union-based method includes all signals, even conflicting or incorrect ones. Intersection outperforming union indicates that precision is currently more critical than recall for LVLMs. In other words, intersection reduces false positives from visual guidance, whereas union increases true positives at the cost of more false positives.
> For example, LLaVA, one of the earliest LVLMs, is particularly prone to hallucinations with complex or noisy instructions and thus benefits greatly from intersection-based methods. Newer models, such as LLaVA-v1.5 and mPLUG-Owl2, are more robust but remain sensitive to partially incorrect inputs.
> ### Q3.2 Why use different prompts in MARINE-intersec and MARINE-union?
> We evaluated MARINE-union using the exact same prompt template as MARINE-intersec, which underperformed. We hypothesize this is because the original version provides more detailed information, whereas reusing the intersec-style prompt combined reliable grounding with potentially misleading false positives, thus weakening the visual guidance.
> | **Model** | **LLaVA** | | **LLaVA-v1.5** | | **mPLUG-Owl2** | |
> |-----|-----|----|-----|------|-----|------|
> | **CHAIR** | $C_S \downarrow$| $C_I \downarrow$ | $C_S \downarrow$ | $C_I \downarrow$ | $C_S \downarrow$ | $C_I \downarrow$ |
> | MARINE-union | **30.4** | 9.7 | **5.4** | **2.7** | **4.8** | **2.7** |
> | MARINE-union (new) | 32.6 | 9.7 | 7.8 | 3.9 | 6.2 | 3.5 |
> *New: Same prompt template as MARINE-intersec.
>
> Note: Greedy and Intersec results are reported in Table 7 of the main paper.
>
> ### Q4. More recent works included related work but not baselines.
> The more recent works were concurrent to our project development, and thus we included them in discussion but not empirical comparison. In the following, we additionally include
>
> ### Q5. Typos
> Thank you for the catch. We will ensure to correct them in our next revision.
>
> ### Q6. What is the "noise intensity" for DETR?
> DETR outputs a confidence score for each object. We filter predictions using a threshold. lower thresholds allow more noisier detections, while higher ones yield fewer, more precise results. This threshold defines the noise intensity.

---

> > ### Comment · Reviewer_SDtF · 2025-04-04
> >
> > Thank you for the additional details!
> >
> > Some follow-up comments:
> >
> > ### Q2. Questions on guidance strength. Is mPLUG-OWL2 an outlier in Figure 3?
> >
> > While I agree that a higher guidance strength γ will potentially lead to worse instruction following (but improved recall and hallucination metrics), I think Figures 9 and 13 are insufficient to motivate the used value of 0.7
> >
> > Ideally, some quantitative metric would be added that more clearly motivates the chosen setting. Alternatively, better highlighting this in the discussion and adding more models to Figure 3 (to see how much of an outlier mPLUG-OWL2 really is), would go some way in this regard.
> >
> > ### Q3.2 Why use different prompts in MARINE-intersec and MARINE-union?
> >
> > Thank you for the additional table. It begs the question how MARINE-intersec would have performed with the "old" prompt.
> >
> > ### Q4. More recent works included related work but not baselines.
> >
> > Your answer seems truncated: `In the following, we additionally include`
> >
> > ### Q6. What is the "noise intensity" for DETR?
> >
> > Then maybe call this "score threshold"? I was not aware that this threshold is called "noise intensity".

---

> > > ### Author Response · Authors · 2025-04-07
> > >
> > > Thank you for your thoughtful comments and suggestions. Please find our responses below:
> > >
> > > ### Q2. Guidance Strength – Is mPLUG-OWL2 an outlier in Figure 3?
> > > We added experiments on LLaVA-v1.5 for Figure 3 and present the numbers below, which shows a similar trend as mPLUG-OWL2: higher γ improves recall but slightly degrades CHAIRs/CHAIRi. This consistency supports γ = 0.7 as a balanced choice. We will update the figure and discussion accordingly in our revision. Thanks again for the helpful suggestion.
> > > | Guidance Strength γ | CHAIRs | CHAIRi | Recall |
> > > |-----------|--------|--------|--------|
> > > | 0.0       | 0.088  | 0.0457 | 0.4114 |
> > > | 0.1       | 0.076  | 0.0382 | 0.4187 |
> > > | 0.2       | 0.074  | 0.0375 | 0.4260 |
> > > | 0.3       | 0.066  | 0.0343 | 0.4333 |
> > > | 0.4       | 0.064  | 0.0311 | 0.4419 |
> > > | 0.5       | 0.058  | 0.0287 | 0.4501 |
> > > | 0.6       | 0.058  | 0.0288 | 0.4647 |
> > > | 0.7       | 0.062  | 0.0300 | 0.4430 |
> > > | 0.8       | 0.050  | 0.0259 | 0.4706 |
> > > | 0.9       | 0.056  | 0.0289 | 0.4779 |
> > > | 1.0       | 0.062  | 0.0325 | 0.4834 |
> > >
> > > ### Q3.2. Why use different prompts in MARINE-intersec and MARINE-union?
> > > Thank you for raising this point. We include MARINE-intersec with the other prompt version used in MARINE-union originally. The updated results are as follows:
> > >
> > >
> > > | **Model**    | **LLaVA** |       | **LLaVA-v1.5** |       | **mPLUG-Owl2** |       |
> > > |--------------|-----------|-------|----------------|-------|----------------|-------|
> > > | **CHAIR**    | $C_S \downarrow$ | $C_I \downarrow$ | $C_S \downarrow$ | $C_I \downarrow$ | $C_S \downarrow$ | $C_I \downarrow$ |
> > > | Greedy       | 26.6      | 10.5  | 8.8            | 4.6   | 6.2            | 3.4   |
> > > | MARINE-Intersec       | 17.8      | 7.2   | 6.2            | 3.0   | 4.2            | 2.3   |
> > > | MARINE-Intersec (*)   | 24.4      | 8.3   | 7.0            | 3.5   | 6.0            | 3.0   |
> > > *: Same prompt template as MARINE-union.
> > >
> > > As shown, the original prompt for MARINE-intersec leads to consistently better CHAIR scores across all models. We will include this comparison in the appendix and clarify our prompt design choice in the main text. We will include this comparison in the updated appendix and clarify our design choices in the main text.
> > >
> > > ### Q4. Related work but no baselines
> > > Apologies for the typo in our previous response. We include below our original response:
> > >
> > > The more recent works were concurrent to our project development, and thus we included them in discussion but not empirical comparison. These works [1-2] share similar intuition of leveraging classifier-free guidance to enhance LVLMs as ours. However, their primary objectives and evaluation benchmarks differ from ours, making direct comparisons not suitable without specific adaptation..
> > >
> > > [1] Prompt Highlighter: Interactive Control for Multi-Modal LLMs
> > >
> > > [2] Contrastive Region Guidance: Improving Grounding in Vision-Language Models without Training
> > >
> > > ### Q6. “Noise intensity” in DETR
> > > Thank you for pointing this out. We will revise the terminology in the next version to avoid confusion.

---

### Official Review · Reviewer_yYZG · 2025-03-17

**Overall Recommendation:** 4

**Summary:**

This paper proposes a framework (MARINE) that aggregates a VLM and traditional vision tools such as object detection and image-text alignment. Concretely, given an input image, MARINE uses vision tools as guidance models, achieved through a linear combination of unconditional and conditional logits over the vocabulary. This framework can adopt VLMs where logits are accessible and vision tools, and it does not require fine-tuning or any processing via an API for the vision tools.

This approach is primarily evaluated using two automatic metrics for hallucination detection in text (CHAIR and POPE), and it’s compared against various decoding methods. On average, across 5 different base VLMs, MARINE outperforms the baselines in terms of CHAIR and POPE metrics, with the exception of CHAIR recall. Additionally, a GPT-4V-based evaluation is conducted (following the LLaVA paper), along with assessments on other vision-language tasks (VQA) and ablation studies.

**Claims And Evidence:**

The main claim of this paper is that using vision tools as guidance effectively and efficiently mitigates hallucination in image-to-text generation. This paper provides evidence for this claim (CHAIR/POPE results and latency analysis). However, I feel the choice of automatic metrics for hallucination detection can be improved (see my comments in “Methods And Evaluation Criteria”).

**Essential References Not Discussed:**

See the “Relation To Broader Scientific Literature” section.

**Experimental Designs Or Analyses:**

See my comments in the “Methods And Evaluation Criteria” section.

**Methods And Evaluation Criteria:**

**Methods**

The proposed approach is clearly explained in this paper, and the motivations behind it seem reasonable. The design choices appear well-justified given the primary goals: flexibility (training/API-free), simplicity, and effectiveness.


**Evaluation**

I feel CHAIR is limited to evaluate VLMs that can generate long and detailed image captions/descriptions. CHAIR uses MSCOCO captions and their object annotations, but these captions are considerably shorter than what current VLMs can generate. POPE relies on a segmentation tool to obtain ground truth objects, and some negative sampling methods could make a single question too easy (e.g., random negatives could be too easy).

Although hallucination detection metrics are beyond the scope of this paper, the scope of the metrics used should be clearly explained. Newer automatic metrics for hallucination such as ALOHa (https://aclanthology.org/2024.naacl-short.30.pdf) and Visual Fact Checker (https://arxiv.org/pdf/2404.19752) have been introduced. Including results from these newer and more robust metrics would strengthen this paper. In addition to automatic metrics, performing human evaluation to verify their robustness would also be helpful.

**Other Comments Or Suggestions:**

See my comments in “Methods And Evaluation Criteria”

**Other Strengths And Weaknesses:**

- Overall, this paper is well-written and easy to follow. Supplementary materials typically provide additional information to clarify ambiguous points in the main text.
- Just to clarify my position, my only concern is the choice of automatic metrics. I had hoped that this paper would explain the limitations and scope of those metrics or use more up-to-date ones.

**Questions For Authors:**

- A.5 says “Besides, we employed the synonym list from Lu et al. (2018) to align synonymous words in the generated text with MSCOCO object categories.” Is this a common practice with CHAIR? If not, how does this affect the final numbers?

**Relation To Broader Scientific Literature:**

A line of research on image-text alignment might be related. It would be nice to mention some key papers as related work (e.g., TIFA, DSG, Gecko, VQAScore). Also, newer metrics for detailed image captions such as CAPTURE (https://arxiv.org/pdf/2405.19092) could be related.

**Theoretical Claims:**

N/A

---

> ### Author Rebuttal · Authors · 2025-04-01
>
> We sincerely appreciate your thoughtful review and encouraging feedback. Thank you for recognizing the clarity and practical design of our approach, our emphasis on mitigating hallucinations, and the strong empirical support we provided. Below, we provide detailed responses to your comments:
>
> ### Q1. Discuss the limitations and the scope of the metrics used in the paper.
> Thank you for your suggestion and for pointing out recent developments in hallucination evaluation. CHAIR and POPE are widely adopted and reliable metrics for evaluating hallucination, and we believe they offer reliable assessments within their respective scopes. Nonetheless, we acknowledge their inherent limitations: CHAIR depends on a predefined list of object classes and synonyms, which may struggle to detect uncommon objects or very nuanced attributes. POPE’s reliability, in turn, can be influenced by the segmentation tool used to obtain ground-truth objects.
> We will include a brief discussion of these limitations in the revised paper and also report results using ALOHa to complement our evaluation.
> ### Q2. Use up-to-date automatic metrics such as ALOHa.
> Thank you for pointing out the newer evaluation metrics. We will include them in discussion. We have incorporated ALOHa into our evaluation to better assess localizable object hallucinations. Specifically, we report both the object-level hallucination score ($ALOHa_0$) and the caption-level aggregated score ($ALOHa$). As shown in the table below, MARINE consistently reduces object hallucinations and outperforms baseline generations across all settings and metrics.
>
> | Method  | LLaVA-$ALOHa$ $\uparrow$     | LLaVA-$ALOHa_0$ $\uparrow$ | LLaVA-v1.5-$ALOHa$ $\uparrow$     | LLaVA-v1.5-$ALOHa_0$ $\uparrow$     | mPLUG-Owl2-$ALOHa$ $\uparrow$     | mPLUG-Owl2-$ALOHa_0$ $\uparrow$     |
> |-------|--------|------|--------|--------|--------|--------|
> | Greedy   | 40.1%  | 70.1%  | 61.9%  | 83.1%   | 70.2%  | 87.0%  |
> | MARINE   | 48.7%$_{+8.6}$   | 76.1%$_{+6.0}$  | 66.7%$_{+4.8}$  | 85.6%$_{+2.5}$  | 72.9%$_{+2.7}$      | 88.2%$_{+1.2}$ |
>
> Note: For implementation details, we use MSCOCO ground-truth captions as references and enable reference object detection for more localizable and generalizable object hallucinations detection.
>
> ### Q2.1. Evaluation in addition to automatic metrics
> Thanks for the suggestion. In Table 3 of our manuscript, we did include GPT-4v evaluation, which compares the outputs of two LVLM assistants using GPT-4V as a judge. This "LLM-as-a-Judge" evaluation protocol has become widely accepted as a reliable proxy for human evaluation, particularly when large-scale human assessments are costly.
>
> Although conducting extensive human evaluations is beyond the scope of our rebuttal timeline, we fully acknowledge its importance. We consider incorporating comprehensive human evaluations as future extensions of this research.
>
> ### Q3. Is using synonym list a common practice with CHAIR?
> Yes, this is standard practice in the original CHAIR paper [1] and its official implementation. Specifically, they use a synonym list from Lu et al. (2018) [2] to map words (e.g., “player”) to MSCOCO object categories (e.g., “person”).
>
> [1] Object Hallucination in Image Captioning
>
> [2] Neural Baby Talk
>
> ### Q4. Suggested related work.
> Thank you for your suggestion on this line of research on faithfulness evaluation for text-to-image generations, parallel to our focus on image-to-text generations. We gave the mentioned research a careful read and summarized them below.
>
> TIFA focuses on evaluating text-to-image generation by automatically generating and answering questions derived from prompts, measuring faithfulness along several categories (objects, actions, attributes). DSG (Davidsonian Scene Graph) also assesses text-to-image alignment but emphasizes a structured decomposition of prompts into atomic propositions. VQAScore introduces a VQA-based metric that better captures compositional semantics (such as object relationships and attributes), providing insights for attribute and relationship level object hallucination evaluation. Finally, Gecko addresses text-embedding tasks: it uses LLMs to synthesize query–passage pairs (and carefully select hard negatives) in order to train a compact yet powerful universal representation model.
>
> CAPTURE introduces a structured metric for evaluating detailed image captions by extracting and aligning objects, attributes, and relations across captions. We will include this in related work discussion in our next revision.

---

> > ### Comment · Reviewer_yYZG · 2025-04-04
> >
> > Thank you for your responses to my questions/concerns. Since my major concerns have been addressed (up-to-date metrics), I updated my assessment.

---

> > > ### Author Response · Authors · 2025-04-07
> > >
> > > Thank you for your encouraging feedback on our rebuttal! We're delighted to hear that we've addressed your main concerns.

---

### Decision · Program_Chairs · 2025-05-01

**Decision:**

Accept (spotlight poster)

**Comment:**

This work proposes a method to reduce hallucinations in large VLMs. The work received initially positive scores (3,3,3,4). The authors submitted a rebuttal, which helped clarify some of the comments provided by the reviewers. Eventually, the reviewers fully leaned towards accepting the work based on the novelty of the idea, the clarity of the presentation and the effectiveness of the provided results. The recommendation from the AC is to strong accept this work for the conference.